# GRF2 Is Crucial for Cone Photoreceptor Viability and Ribbon Synapse Formation in the Mouse Retina

**DOI:** 10.3390/cells12212574

**Published:** 2023-11-04

**Authors:** David Jimeno, Concepción Lillo, Pedro de la Villa, Nuria Calzada, Eugenio Santos, Alberto Fernández-Medarde

**Affiliations:** 1Centro de Investigación del Cáncer-Instituto de Biologıá Molecular y Celular del Cáncer (CSIC–Universidad de Salamanca) and CIBERONC, 37007 Salamanca, Spain; 2INCYL, IBSAL (Universidad de Salamanca), 37006 Salamanca, Spain; 3Departamento de Biología de Sistemas, Universidad de Alcalá, 28871 Alcalá de Henares, and IRYCIS, 28034 Madrid, Spain

**Keywords:** cone photoreceptor, GRF2, CDC42, RAC1, nuclear movement, cell death, ribbon synapsis, retina, degeneration

## Abstract

Using constitutive GRF1/2 knockout mice, we showed previously that GRF2 is a key regulator of nuclear migration in retinal cone photoreceptors. To evaluate the functional relevance of that cellular process for two putative targets of the GEF activity of GRF2 (RAC1 and CDC42), here we compared the structural and functional retinal phenotypes resulting from conditional targeting of RAC1 or CDC42 in the cone photoreceptors of constitutive GRF2^KO^ and GRF2^WT^ mice. We observed that single RAC1 disruption did not cause any obvious morphological or physiological changes in the retinas of GRF2^WT^ mice, and did not modify either the phenotypic alterations previously described in the retinal photoreceptor layer of GRF2^KO^ mice. In contrast, the single ablation of CDC42 in the cone photoreceptors of GRF2^WT^ mice resulted in clear alterations of nuclear movement that, unlike those of the GRF2^KO^ retinas, were not accompanied by electrophysiological defects or slow, progressive cone cell degeneration. On the other hand, the concomitant disruption of GRF2 and CDC42 in the cone photoreceptors resulted, somewhat surprisingly, in a normalized pattern of nuclear positioning/movement, similar to that physiologically observed in GRF2^WT^ mice, along with worsened patterns of electrophysiological responses and faster rates of cell death/disappearance than those previously recorded in single GRF2^KO^ cone cells. Interestingly, the increased rates of cone cell apoptosis/death observed in single GRF2^KO^ and double-knockout GRF2^KO^/CDC42^KO^ retinas correlated with the electron microscopic detection of significant ultrastructural alterations (flattening) of their retinal ribbon synapses that were not otherwise observed at all in single-knockout CDC42^KO^ retinas. Our observations identify GRF2 and CDC42 (but not RAC1) as key regulators of retinal processes controlling cone photoreceptor nuclear positioning and survival, and support the notion of GRF2 loss-of-function mutations as potential drivers of cone retinal dystrophies.

## 1. Introduction

The GRF family of guanine nucleotide exchange factors (GEFs) encompasses two highly homologous members, GRF1 and GRF2, which are preferentially expressed in the central nervous system (CNS) and capable of activating different members of the RAS or RHO/RAC subfamilies of small GTPases in a variety of mammalian cell types. Despite their homology, a variety of distinct, differential functionalities has been ascribed to GRF1 and GRF2 in different biological contexts [1]. Thus, our previous analyses of GRF1/2 KO mice revealed specific functional roles of GRF1 in pancreatic beta cells [2] and neurosensory and photoreception processes [3,4]. Likewise, we also uncovered critical functional contributions of GRF2 regarding substance addiction behavior [5], the control of stem cell density and differentiation during adult neurogenesis [6], and in the correct development and function of the cone photoreceptors that are essential for correct color vision in the postnatal mouse retina [7,8].

During postnatal retinal development, the nuclei of neuronal progenitor cells undergo the so-called interkinetic nuclear migration (INM) by which the nuclei of precursor cells initially move perpendicularly to the neuroepithelium [9]. This nuclear migration is coupled to the cell cycle in such a way that the cells reach the M phase when their nuclei are in the apical side of the neuroepithelium [10]. Later in retinal development, cone nuclei move to the apical side of the ONL (outer nuclear layer), where they are finally located in the adult retina [11]. Although the precise significance of this movement is not clearly understood, defects in migration have been linked to abnormal retinal development [12,13]. In addition, normal nucleus positioning is needed for a correct establishment of synaptic terminals, or even for cone cell survival [14,15,16]. In humans, the mispositioning of cone photoreceptor nuclei increases in an age-dependent manner [17] and this mislocalization is further accelerated in retinas affected by age-related macular degeneration [18].

The structural cellular elements involved in nuclear migration are well known. LINC (linker of nucleoskeleton and cytoskeleton) complexes are formed by scaffold proteins that span the nuclear envelope (NE) and connect the nuclei to the cytoskeleton [14]. Within the LINC complex, the SUN proteins are known to interact with lamins, and the nesprins bind to the SUN proteins through their KASH domains and to different cytoskeletal elements through their cytoplasmic region [19,20,21,22]. Different approaches, designed to disrupt components of the LINC complex, have demonstrated the importance of these proteins in the final nuclear positioning in adult cone photoreceptors [15,23,24].

Some of the signaling molecules and regulatory elements that participate in nuclear movements have been described in myocytes and in the granule cells of the cerebellum. CDC42 and its signaling partners, Par3, Par6, and aPKC, have been shown to play a crucial role in nuclear migration processes in these systems [25,26,27]. Moreover, our laboratory has shown that disrupting GRF2 expression has a significantly deleterious impact on the location of cone nuclei in the retina that is also associated with a defective light perception by the cones [7,8]. Since a direct, functionally productive interaction between CDC42 and GRF2 has been reported [28,29], we investigated here if the defects of nuclear translocation seen in GRF2^KO^ mice might be caused by the disruption of this interaction between GRF2 and CDC42. Furthermore, as GRF2 is also an activator of RAC1, a critical player in actin dynamics [30,31], and the actin cytoskeleton is an anchor for nuclear movement [32], we also investigated in this report the role of RAC1 in cone nuclear movement and its potential implication in the defects observed in the GRF2^KO^ retina. Our data underscore the critical mechanistic contribution of GRF2 and CDC42, but not RAC1, to the process of the control of nuclear migration required for correct development, as well as for normal function and neural connectivity, of the retinal cone photoreceptors.

## 2. Materials and Methods

### 2.1. Animals

Animal protocols in this study were carried out according to the European Communities Council Directive of 20 March 2015 (ECC/566/2015), and Spanish guidelines (RD53/2015) for the use and care of animals in research. The experiments were designed to comply with the three R’s policy and were approved by the University of Salamanca Animal Welfare Committee and Castilla y León regional government (Approval # 413).

All genetically modified mouse (GEMM) strains used in this study were previously backcrossed at least 6 times with C57Bl/6J wild-type mice to ensure a homogeneous genetic background in all experiments. All animals were maintained under a 12 h light/12 h dark cycle and fed ad libitum. Constitutive null GRF2^KO^ mice, floxed RAC1 mice, floxed CDC42 mice, and transgenic mice expressing CRE recombinase in cone photoreceptor cells under control of the human red/green pigment (HRGP) gene were previously described [33,34,35,36]. Appropriate cross-mating among these mouse lines was used to generated the different relevant genotypes analyzed in this study, including GRF2^−/−^/HRPG-CRE^+/−^ (hereafter, GRF2^KO^); CDC42fl/fl/HRPG-CRE^+/−^ (hereafter, CDC42^KO^); RAC1fl/fl/HRPG-CRE^+/−^ (hereafter, RAC1^KO^); GRF2^KO^/CDC42fl/fl/HRPG-CRE^+/−^ (hereafter, GRF2^KO^/CDC42^KO^); and GRF2^KO^/RAC1fl/fl/HRPG-CRE^+/−^ (hereafter, GRF2^KO^/RAC1^KO^). WT mice maintained in the same genetic background were used as controls throughout the study.

### 2.2. Immunohistochemistry

Animals anesthetized with isoflurane were euthanized via cervical dislocation. A nick was used to mark the temporal part of the retina, before removing and fixing the eyes in 4% formaldehyde for 4 h or 2 h at 4 °C in 4% formaldehyde as indicated. The fixed samples were washed with PBS before removing the cornea and the lens. The eyes were then cryoprotected via successive immersions in 10% and 30% sucrose, before embedding in TissueTek and freezing in liquid nitrogen. Then, 16 µm dorso-ventral sections were obtained in a cryostat and stored at −80 °C until used. The sections were washed with PBS and blocked with 0.1% Triton X-100, 5% BSA, and 2% goat serum (GS) in PBS for 1 h at room temperature (RT). Primary antibodies (Table 1) were diluted in PBS with 0.1% Triton X-100, 2% BSA, and 2% GS, and incubated overnight at 4 °C. Lectin from Arachis hypogaea-FITC (PNA-FITC) was incubated for 1 h at RT. After three washes in PBS, the samples were incubated with the secondary antibodies (Jackson ImmunoResearch, West Grove, PA, USA) either goat anti-rabbit Alexa Fluor 488 or Cy3 or goat anti-mouse Alexa Fluor 488 or Cy3, diluted at 1:600 for 1 h at RT—washed with PBS, and mounted with ProLong Gold anti-fading reagent (Life Technologies, Carlsbad, CA, USA). When appropriate, the nuclei were counterstained with Hoechst 33,342 (Life Technologies). For pMLC2 quantification, images were imported to ImageJ software v.1.53t (NIH, Bethesda, MD, USA), the background was corrected, and raw fluorescence was quantitated in three independent images from at least three different animals.

### 2.3. Electron Microscopy

The eyes were marked in the temporal area before fixing in 2% formaldehyde and 2% glutaraldehyde for at least 24 h at 4 °C. Then, the anterior portion of the eyeballs was removed, and the eyecups were post-fixed in 1% aqueous osmium tetroxide, dehydrated in ethanol, and embedded in Embed 812 resin (Electron Microscopy Sciences, Hatfield, PA, USA). Ultrathin sections (50 nm) were prepared using a ultramicrotome Ultracut E (Leica, Wetzlar, Germany) and mounted on copper grids, stained with uranyl acetate and lead citrate, and analyzed with a Tecnai Spirit Twin 120 kv transmission electron microscope. Micrographs were obtained using a coupled digital camera (CCD Gatan ES1000W (Pleasanton, CA, USA)) and ImageSP software (Sysprog, Minsk, Belarus).

### 2.4. BaseScope In Situ Hybridization (ISH)

To detect GRF2 mRNA in the retinal tissue sections, we used the ACD BaseScope™ Reagent Kit-RED ISH system following the manufacturer’s instructions. In brief, the eyes were fixed in 10% neutral buffered formalin at RT for 24 h and embedded in paraffin. Five-micron sections were collected on Superfrost Plus slides and we followed the user manual instructions, performing 15 min of antigen retrieval and 30 min of protease III treatment. A specific GRF2 probe was designed, covering the exon junction area, including exons 8-9 (BaseScope™ Probe-BA-Mm-Rasgrf2-2EJ, (ACDboi, Newark, CA, USA), Cat#711291). Since the GRF2^KO^ mice were generated eliminating exon 9 (lacking the area detected with the probe), we used our GRF2^KO^ mice as negative controls.

### 2.5. Quantification of Cone Photoreceptor Nuclear Positioning

An analysis of the cone nuclei distribution throughout the outer nuclear layer (ONL) was performed as previously described [7]. Briefly, retinas of 15 days of age of all genotypes analyzed were immunostained with β-catenin and red/green opsin. From each retina, three sections corresponding to the nasal, central, and temporal areas were analyzed. Confocal images of the dorsal and ventral parts of the retina were obtained. Images were imported into the ImageJ software (NIH) and the distances from the cone cell body (labeled with red/green opsin antibody), to the outer limiting membrane (OLM) (labeled with the β-catenin antibody) and total thickness of the ONL in that area were measured. The relative position of each cone nucleus in the ONL was calculated, with 0 being the position closest to the OLM and 10 being the inner part of the ONL. The ONL was divided into 10 equal regions and each cone nucleus was assigned to one of these regions according to its relative position in the ONL.

### 2.6. Cone Cell Number Quantification

To quantify the number of cone cells in the retinas of the different genotypes analyzed, we took advantage of the strong and stable expression of CRE recombinase in the cones of HRPG-CRE^+/−^ animals and identified the CRE^+^ cells using immunohistochemistry, as described above. Overlapping images were collected using a DeltaVision microscope with a 20× objective covering the whole dorso-ventral retina section. Images were imported into the ImageJ software, converted to jpg format, and a 2D reconstruction of the whole section was performed using the grid/collection stitching plugin [37]. We counted the total number of CRE-positive nuclei in the ONL of the whole retina sections, and to correct for small differences in the total section length, we calculated the number of cone cells/mm of the retinal sections.

To evaluate the differences in the cone cell numbers in the different retinal regions, we performed a separated quantification of CRE^+^ cells in different areas of the retina. The dorso-ventral retinal sections were used and the number of CRE^+^ nuclei was counted in a 100 μm stretch of each retina section every 250 μm, starting in the optic nerve and heading towards the dorsal and ventral parts of retina.

### 2.7. Statistical Analysis

A minimum of three sections for each retina were counted for each experimental procedure. The specific number of animals used for individual experiments can be found in the figure legends. The statistical method used in this study was Student’s *t*-test with Bonferroni post hoc.

### 2.8. TUNEL

The apoptotic rates in the mice retinas were analyzed using a Click-iT^®^ Plus TUNEL Assay with an Alexa Fluor^®^ 488 dye for the “In Situ” apoptosis detection of double-strand breaks, following the manufacturer’s instructions (Thermo Fisher Scientific (Waltham, MA, USA), Cat# C10617).

### 2.9. Electroretinogram Recordings

ERG assays were performed as previously described [38]. Briefly, the mice were adapted to darkness overnight and ERG responses were flash-induced with a Ganzfeld stimulator, recorded, and analyzed. The rod, mixed rod, and cone responses were measured in the WT (*n* = 7), CDC42^KO^ (*n* = 9), GRF2^KO^ (*n* = 8), and GRF2^KO^/CDC42^KO^ (*n* = 7) animals.

## 3. Results

### 3.1. Spatio-Temporal Pattern of GRF2 Expression in the Mouse Retina

Although we have previously shown the expression of GRF1 and GRF2 at the protein level during the postnatal development of the retina [7], it is widely recognized that the currently available antibodies lack sufficient specificity for accurate immunohistological localization studies of these two structurally related GEFs. Hence, in order to enable a precise, specific detection of GRF2 mRNA expression in the different layers/cell types of the retina, here, we performed in situ hybridization assays using specific probes capable of easily discriminating between potential, partially homologous GRF1 and GRF2 gene products (see Section 2). Thus, consistent with our prior observations via a Western blot analysis [7], we verified that GRF2 mRNA is widely expressed postnatally in the retina of WT mice (Figure 1). In particular, we identified a distinct peak of GRF2 expression at ages P7 and P15 (Figure 1B–C), a temporal window coinciding with the developmental stage at which the nuclei of the cones are physiologically moving toward the apical part of the ONL, close to the OLM [9,10], and also matching the timing of our previously described peak of the detection of ectopic nuclei in GRF2^KO^ retinas [7]. Our in situ assays also showed that GRF2 mRNA expression was maintained, although at more reduced levels, at later ages in the adult WT mice (Figure 1D,E). As a negative control, our assays showed a complete absence of GRF2 signals in the retinas of the GRF2^KO^ mice (Figure 1F). Regarding the outer retina, GRF2 is expressed in both photoreceptor cell types as well as in the Muller cells of WT mice, as shown through the co-localization of the GRF2 probes with peanut agglutinin (PNA) and the glutamate–aspartate transporter (GLAST), respectively (Appendix A).

### 3.2. GRF2 and CDC42 (but Not RAC1) Participate in Cone Nuclear Translocation

To quantitatively evaluate the implication of two potential GRF2 targets, CDC42 and RAC1, in the process of nuclear migration in cone photoreceptors, we measured the actual distance of the cone nuclei to the OLM at P15 (a stage when the migration has already ended) in retinal sections from mice of the six relevant genotypes analyzed in this study (Figure 2). After identifying the cone nuclei with anti-R/G opsin (red) and the OLM with β-catenin (green) immunostaining (Figure 2A), the distance from each nucleus to the OML was measured and the mean distances were calculated and plotted for each genotype (Figure 2B).

Consistent with our previous report [7], the cone nuclei of the GRF2^KO^ mice aligned closer to the OLM than in the WT animals, even surpassing this limit at times and entering into the photoreceptor segments (PS) layer [7] (Figure 2). In sharp contrast, our analysis of cone nuclei in the CDC42^KO^ retinas revealed that they were spread throughout the outer nuclear layer (ONL) and did not accumulate in the proximity of the OLM, which physiologically happens in the WT retinas (Figure 2). These contrasting nuclear positioning data suggest that GRF2 and CDC42 may play opposite regulatory roles in the control of the process of cone nuclear migration, in accordance with previous reports showing mutually inhibitory functional interactions between these two signaling molecules in other biological contexts [28,29,39,40]. To confirm this view, we crossed GRF2^KO^ animals with CDC42-CRE null mice and analyzed the nuclear movement in the resulting GRF2^KO^/CDC42^KO^ cone photoreceptors. These experiments showed that these double-knockout (DKO) retinas recovered a normal cone nuclear positioning pattern, with the nuclei aligning close to the OLM layer as in the WT retinas (Figure 2A,B), further supporting the notion of opposed functional roles for GRF2 and CDC42 in the process of cone nuclear migration. Interestingly, the single-knockout RAC1^KO^ retinas displayed a pattern of cone nuclear migration completely similar to that of normal WT retinas, whereas the double-knockout GRF2^KO^/RAC1^KO^ retinas showed a nuclear migratory phenotype that was undistinguishable from that of the single GRF2^KO^, suggesting that the defects in nuclei migration in the GRF2^KO^ mice are not mechanistically related to a lack of RAC1 activation, and that RAC1 is not implicated in the regulation of cone nuclear movement in mice (Figure 2).

To better understand the defects in nuclear positioning, we analyzed the position of the nuclei inside the ONL region. Figure 2C presents a quantitative analysis of the data on nuclear distribution through ten defined areas of the ONL region in the retinas of the P15 mice of the relevant GRF2, CDC42, or RAC1 genotypes. Our measurements showed that the distribution of cone nuclei in the WT, RAC1^KO^, and GRF2^KO^/CDC42^KO^ retinas was almost identical in all cases, with a majority of cone nuclei accumulating in the second uppermost area of the ONL (Figure 2C). In contrast, the GRF2^KO^ and GRF2^KO^/RAC1^KO^ retinas showed a similar high percentage of nuclei located in the top layer, abutting the OLM (ectopic nuclei above the OLM were excluded from these quantifications). Remarkably, the location of the cone nuclei in the retinas of the single CDC42^KO^ mice was evenly spread throughout the ONL region, confirming a defect in nuclear movement in these mice.

### 3.3. Opposite Effects of RAC1 and CDC42 Ablation on GRF2-Dependent MLC2 Phosphorylation in the Mouse Retina

It was previously shown that GRF2 suppresses the CDC42-dependent phosphorylation and activation of MLC2 (Myosin Light Chain 2) in cultured cell lines from different human tumor types [28]. Consistent with that report, we also reported that GRF2 ablation is accompanied by specifically increased levels of activated, phosphorylated pMLC2 in the mouse retina [7], in synchrony with the excessive nuclear movement occurring in GRF2^KO^ cone photoreceptors [7]. In this regard, we wished to search for additional mechanistic clues by analyzing the status of retinal MLC2 phosphorylation after the specific ablation of RAC1 or CDC42 via the means of immunoassays using anti-phospho-MLC2 antibodies (Figure 3).

Interestingly, the immunostaining and quantification of the pMLC2 signal showed that the single elimination of CDC42 resulted in increased levels of activated, phosphorylated MLC2 (pMLC2) in the retina in comparison to their WT control counterparts, and furthermore, that the pMLC2 levels were even higher in the double-knockout retinas lacking both GRF2 and CDC42 (Figure 3). Given this synergistically negative effect of CDC42 and GRF2 ablation on MLC2 activation in the retina, the opposite impact on cone nuclear movement observed in single-knockout CDC42^KO^ and GRF2^KO^ retinas, together with the correction of those alterations of nuclear migration in the double-knockout CDC42^KO^/GRF2^KO^ retinas (Figure 2), suggest that the process of MLC2 phosphorylation may be, at least in part, mechanistically unrelated or independent from the process of nuclear migration.

On the other hand, although not reaching statistical significance, we observed a trend suggesting that RAC1 elimination in the retina leads to a reduction in MLC2 phosphorylation in comparison to their RAC1^WT^ controls (Figure 3).

### 3.4. Progressive Cone Photoreceptor Loss Caused by GRF2 Ablation Is Aggravated by Concomitant CDC42 (but Not RAC1) Ablation

Since alterations in nuclear movement have been previously linked to various physiological alterations involving the degeneration of the cone photoreceptors [18,41], we wished to analyze if the absence of GRF2 and/or the RAC1 or CDC42 GTPases could also alter/impact the rate of cellular survival of the cones. To that aim, we performed detailed measurements of the total count of cone photoreceptors present at different postnatal stages in the retinas of our different KO mouse strains (all of them kept on an HRPG-CRE^+/−^ background) via the means of the immunohistochemical detection of the CRE protein specifically expressed in the cones under the control of the HPRG promoter (Figure 4).

Our observations at P15 and 5 months of age showed that the single ablation of either RAC1 or CDC42 resulted in similar counts of retinal cone photoreceptors than in the retinas of similarly aged WT mice (Figure 4A). On the other hand, our analysis showed a significant reduction in the number of cones counted in the retinas of the 5-month-old GRF2^KO^ mice when compared to the WT controls of the same age or to the retinas of the P15 mice of the same GRF2^KO^ genotype (Figure 4A), supporting a causal mechanistic link between the absence of GRF2 and the disappearance of cone photoreceptors in the adult 5-month-old retinas. Interestingly, the combined removal of RAC1 and GRF2 did not modify the pattern of reduced cone count already seen in single GRF2^KO^ retinas (Figure 4A), suggesting that RAC1 does not participate in the mechanisms controlling this process of cone disappearance. In contrast, despite the fact that single CDC42 ablation did not modify the cone cell count by itself (Figure 4A), the reduction in cone numbers caused by single GRF2 ablation was further aggravated when GRF2 elimination was combined with CDC42 removal. Indeed, the double-knockout GRF2^KO^/CDC42^KO^ animals showed an early, statistically significant reduction in cone cell number as soon as P15, and almost all (about 75%) cones were already absent from the retinas after only two additional weeks in the 1-month-old retinas (Figure 4A), suggesting that CDC42 has the capability to participate, at least indirectly, in the process of GRF2-dependent cone photoreceptor loss. The importance of both proteins in cone maintenance is further confirmed by an almost complete absence of cones at 5 months of age in the DKO retinas. In this regard, it was also clearly apparent that the massive reduction in cone numbers occurring in the GRF2^KO^/CDC42^KO^ retinas is mechanistically linked to a substantial increase in the cell death rates of the cones occurring at the early stages of postnatal development, as shown by the high rates of cellular apoptosis already detectable as soon as P23 via the TUNEL assays (Figure 4B).

Finally, a detailed analysis and comparison of the number of cone photoreceptors distributed, at different postnatal ages (P15, P23, 1M, and 3M), throughout the different dorsal and ventral regions of WT, single GRF2^KO^, and double GRF2^KO^/CDC42^KO^ retinas (Figure 4C) uncovered that, while in GRF2^KO^ animals, the cones start dying at later stages and disappear preferentially from the dorsal regions of the retina, the process of cone death/loss was faster and more homogeneously distributed between the dorsal and ventral areas in the GRF2^KO^/CDC42^KO^ retinas (Figure 4C).

Overall, the contrasting retinal phenotypes resulting from the single or combined ablation of GRF2 and/or CDC42 with regard to cone nuclear movement/positioning (Figure 2) or cone survival (Figure 4) underscore/highlight the critical role of GRF2 in the control of cone cell homeostasis and maintenance, and indicate that GRF2 and CDC42 play separate mechanistic roles in the control of the processes of cone nuclear movements and cone cell survival, with both proteins having opposed roles in the control of nuclear movement but synergistic effects in the control of cone cell survival.

### 3.5. Single or Combined Elimination of CDC42 and GRF2 Leads to VASP Hyperactivation

The vasodilator-stimulated protein (VASP) participates in regulation of cell migration and adhesion [42] and is also implicated in cone cell death, observed in the cpfl1 mouse model [41]. Since we previously described that GRF2 elimination results in altered, increased patterns of VASP phosphorylation in the mouse retina [7], we also wished to analyze here the levels of activated pVASP in our CDC42^KO^ retinas (Figure 5).

Comparing to normal WT retinas revealed an overall increase in pVASP expression in the single CDC42^KO^ retinas, and also showed that this increase in VASP activation was even stronger in retinas concomitantly lacking both CDC42 and GRF2 (Figure 5), where many cones in the ONL layer showed similarly high pVASP immunostaining, as previously reported, in the retinas of our single-knockout GRF2^KO^ mice [7].

The fact that pVASP activation (Figure 5) occurred in the single CDC42^KO^ retinas (showing altered cone nuclear positioning) and is also observed in the CDC42^KO^/GRF2^KO^ retinas (where normal cone nuclear movement is restored) suggests that, although GRF2 and CDC42 can modulate pVASP phosphorylation and activation, this process may be functionally unrelated to either cone nuclear movement (double-knockout CDC42^KO^/GRF2^KO^ retinas show normal nuclear movement but higher pVASP activation) or to cone apoptotic cell death/loss (single-knockout CDC42^KO^ retinas do not show cone cell death but exhibit higher pVASP phosphorylation).

### 3.6. Functional Impairment of Cone Photoreceptor Cells in GRF2^KO^ Retinas Is Aggravated in Double-Knockout GRF2^KO^/CDC42^KO^ Mice

We previously showed that the elimination of GRF2 in the mouse retina induces structural alterations leading to defective light perception [7]. Since our double-knockout GRF2^KO^/CDC42^KO^ retinas show normal cone nuclear positioning but significantly high rates of apoptotic death during the first month of postnatal development, we hypothesized that, as the cone cell death rate is faster in the DKO animals, these mice should also present consistent, significant defects in cone-based light perception.

To test this hypothesis, we performed electroretinographic analyses of the DKO animals comparing them to the WT and single GRF2^KO^ mice at 1 month of age (Figure 6). As expected, the b-phot and flicker electroretinographic readings corresponding to light perception in the cones were strongly affected in the double-knockout GRF2^KO^/CDC42^KO^ mice, which showed significantly worsened color and strong light perception than their counterpart GRF2^KO^ animals (b-phot, flicker, Figure 6). On the other hand, the single-knockout CDC42 mice did not show significant alterations in cone functionality in comparison to the WT control mice, and the b-scot recordings of low light perception, a measurement of rod function, was similar for all genotypes tested here (Figure 6).

Our data show that, regardless of the position of the nuclei in the cones, the elimination of GRF2 leads to cone cell death and impaired cone-related light perception, and these functional defects are significantly aggravated after the concomitant elimination of CDC42, despite the fact that such a simultaneous ablation of GRF2/CDC42 reverts the abnormal distribution of the nuclei occurring in the single GRF2^KO^ retinas. All in all, these contrasting retinal phenotypes reinforce the notion that the control of the nuclear positioning/distribution and the apoptotic death rate in retinal cone photoreceptors are probably independent/unrelated cellular events, although they may still be both individually dependent on the expression of GRF2 and CDC42 in this specific retinal cell type.

### 3.7. Specific Structural Alterations in the Synapses of GRF2^KO^ Cone Photoreceptors

As shown above, GRF2 elimination causes defective cone-dependent light perception in the mouse retina. This defective phenotype is probably not related to the location of the nuclei in the cones (a double genetic ablation CDC42^KO^/GRF2^KO^ rescues this defective GRF2^KO^-linked phenotype, reverting it to normal cone nuclear positioning but still showing impaired light perception) but is instead likely due to a progressive degeneration and death of the cones in the retina.

To try and obtain mechanistic clues about the origin of such cone cell death and defective functions, we decided to analyze the cone synapses of our knockout animal strains. For this purpose, we first performed immunostaining assays against peanut agglutinin (PNA, a specific marker for cone photoreceptors [43]) and against 14-3-3γ (a signaling protein localizing to the synaptic connections [44,45]), focusing in particular on the immunostaining patterns observed in the outer plexiform layer (OPL) of the retina (Figure 7). Interestingly, our 14-3-3γ immunoassays detected marked structural defects in the pattern of immunostaining for this protein in the OPL layer of the GRF2^KO^ and GRF2^KO^/CDC42^KO^ retinas. Specifically, the alignment of 14-3-3γ staining in the cone terminals at the OPL layer was strongly altered in the GRF2^KO^ and DKO retinas as compared to the WT controls (Figure 7 insets, arrows), suggesting a specific link between these knockout genotypes and the occurrence of structural alterations in the retinal cone synapses.

To further characterize the structure of the cone photoreceptor synapses in the GRF2^KO^ and CDC42^KO^ retinas, we performed ultrastructural electron microscopy analyses of the same retinal areas in our KO mouse strains (Figure 8). We observed that, while the ribbon synapses in the WT or CDC42^KO^ retinas showed the typical structure surrounded by the membrane invaginations of the postsynaptic elements, the cones in the GRF2^KO^ and GRF2^KO^/CDC42^KO^ retinas presented/exhibited defective, “flattened” synapses, where the ribbon was located in close proximity to a flat portion of the membrane (Figure 8, arrows). These synapses are developed between postnatal days 12 and 30, a stage in which GRF2 is crucial for cone nuclear movement and survival, and are characterized by the accumulation of synaptic vesicles [46,47]. Thus, a reduction in these structures could be responsible for the light perception deficits observed in these animals. In addition, different reports have shown a correlation between abnormal ribbon synapses and cone cell death [48,49]. Also, these alterations in the ribbon synapses are not found in the CDC42^KO^ retinas, in which no cone death is observed (Figure 8). Taken together, these data strongly imply that GRF2 elimination leads to the disruption of the normal synaptic interactions in the cones, subsequent cell death, and defective photoreception.

## 4. Discussion

The experimental observations described in this report clearly indicate that the individual genetic ablation of GRF2 or CDC42 in the mouse retina causes opposite functional effects on the physiological processes controlling/regulating cone photoreceptor nuclear movements, whereas the single ablation of RAC1 does not show any functional impact on those processes.

To try and understand the mechanistic details underlying the opposing mechanistic roles exhibited by GRF2 or CDC42 ablation in the regulation of these processes, we should first take into consideration/account the current state of knowledge about functional interactions between these signaling molecules in different published studies about the regulation of cell polarity in mammalian cells [28,29,50]. In particular, it is known that GRF2 suppresses CDC42-mediated cell movements and cytoskeletal dynamics in tumor cells [28] In this regard, it has been shown that GRF2 can bind to 14-3-3 proteins, while 14-3-3 proteins can bind and increase CDC42 activity [50]. Other studies have also shown the ability of GRF2 to form complexes with GM130 at the Golgi, leading to CDC42 inhibition [29]. Consistent with those reports, our current observations in mouse retinas confirm that functional interactions between GRF2 and CDC42 may play critical regulatory roles in the control of cell polarity and nuclear movement in the cone photoreceptors, although the reverted phenotype observed in the double KO GRF2^KO^/CDC42^KO^ retinas points to more complex mechanisms than the ones described in those early studies, where the elimination of CDC42 would take priority over GRF2 removal [29]. Regarding these mechanisms, it should also be noticed that the single GRF2^KO^ retinas and the double-knockout GRF2^KO^/CDC42^KO^ retinas showed increased levels of pMLC2 activation, whereas the single RAC1^KO^ retinas exhibited clearly reduced MLC2 phosphorylation, suggesting that GRF2 and/or CDC42 might be blocking a signaling pathway linking RAC1 to MLC2 activation. In any event, the process of MLC2 phosphorylation is probably mechanistically unrelated to the regulation of cone nuclear movement, since the RAC1^KO^ animals did not show any alterations in cone nuclear movements, and the co-deletion of GRF2 and RAC in the GRF2^KO^/RAC1^KO^ did not solve nor aggravate the problems observed in the GRF2^KO^ retinas (Figure 2 and [7]).

While our experimental observations appear to clearly exclude a direct functional participation of RAC1 in the control of nuclear positioning in the cone photoreceptors, we are further proposing here a mechanistic explanation (Figure 9) which is consistent with the known suppressing effect of GRF2 over various CDC42-mediated cellular processes [28,29,50], and with the somewhat counterintuitive effects on cone nuclear positioning resulting from the individual or combined ablation of GRF2 and CDC42 (Figure 2 and Figure 4)**.** Our model would not imply a direct negative interaction of GRF2 over CDC42, but rather a more complex suppressing/balancing mechanism whereby the action of GRF2 on RAS proteins would suppress nuclear movement, while the activation of CDC42 by its specific cellular GEFs would play a positive balancing, stimulating effect on those nuclear movements. In this context, the ablation of both GRF2 and CDC42 in DKO retinas is associated with normal cone nuclear movement, implying that additional (still unknown) cellular factors must also be involved in the regulation of nuclear movements in the retinal cone photoreceptors.

Another very relevant phenotype in our studies of KO retinas concerns our detection of significantly increased rates of cellular death and loss/disappearances of cone photoreceptors from the retinas of our GRF2^KO^ and GRF2^KO^/CDC42^KO^ animals (see Figure 2 and Figure 4). It should be mentioned that this specific defective, GRF2-linked phenotype went unnoticed in our previous characterization of GRF2^KO^ retinas of constitutive GRF2^KO^ mice, because in that report, we only counted the total number of photoreceptors, compared to the specific cone count used in our present study. Indeed, unlike prior IHC assays [7], here, we have taken advantage of the CRE expression specifically occurring in the cone photoreceptor cells of our KO mouse strains, so as to allow for much higher sensitivity and the precise quantification of the number of the retinal cone populations present at different stages and ages in the retinas of the different mouse genotypes under analysis. Regarding the underlying mechanism(s) responsible for these phenotypes, in contrast to a previous report suggesting a potential connection between the processes of nuclear movement/positioning and cell death in cone photoreceptors [13], our current experimental model strongly suggests that the mechanisms controlling the process of cone death and loss from the retina are independent from the processes controlling nuclear positioning in those cells, since the CDC42^KO^ retinas show a clearly defective nuclear positioning but not cell death, whereas the GRF2^KO^/CDC42^KO^ retinas shows cone nuclear positioning comparable to the WT, while still undergoing rapid cone cell degeneration (Figure 4). Furthermore, it is also relevant to mention here the very differing kinetics of cone photoreceptor death/disappearance exhibited by single GRF2^KO^ retinas as compared to DKO GRF2^KO^/CDC42^KO^ retinas, thus supporting the participation and interaction of GRF2 and CDC42 in the generation of this phenotype. Indeed, whereas most cones are already dead and disappeared from the DKO GRF2^KO^/CDC42^KO^ retinas at 1 month of age, the rate of cone cell death is much slower in the GRF2^KO^ retinas, with some cones are still present in the retinas of 5-month-old mice. Furthermore, besides the slower rate of cone degeneration, the disappearance of the cones in the GRF2^KO^ retinas was not homogeneous throughout the retina, since cones in the dorsal part of the retina died faster than those in the ventral part (Figure 4). Interestingly, a slow, non-homogeneous cone cell degeneration has also been reported in a number of human retinal degenerative illnesses [51].

Cone photoreceptor cell degeneration is observed in various human retinal degenerative disorders, resulting from a process where this deterioration is sometimes associated with the previous death of rod photoreceptors (secondary cone cell death), while in other instances, is an independent event [52,53]. In this regard, it is relevant to mention that the elimination of either GRF2 [7], CDC42, or both in our mouse KO system leads to higher VASP phosphorylation (Figure 6), a modification that has been linked to cone cell death in the cpfl1 mouse model of cone degeneration [41]. However, the absence of cone cell death in the CDC42^KO^ retinas suggests that VASP activation is not linked to cone apoptosis in our mouse models.

Cone photoreceptor cell death has also been frequently associated with defective synapses in many human degenerative diseases of the retina [52,54]. These observations in humans strongly suggest that the progressive cone photoreceptor death observed in our GRF2 and GRF2^KO^/CDC42^KO^ mice is also a consequence of the defects in the structure of ribbon synapses observed in those mice. We have also previously observed that GRF2 is implicated in adult neurogenesis [6], and other studies have shown its role in the regulation of LTP [55], but, to our knowledge, the role of GRF2 in synapse generation or maintenance is unknown. In this regard, the ability of GRF2 to bind 14-3-3 proteins [50] may play a role in the regulation of synapse formation. In this report, we have shown that 14-3-3γ distribution in the synaptic cleft of the OPL is altered in the GRF2^KO^ retinas. Among the 14-3-3 protein family, 14-3-3γ shows a weak expression in the retina, but is specifically localized to the synaptic region of the OPL layer [45]. In addition, 14-3-3 proteins bind to Bassoon in a phosphorylation-dependent mode, and this interaction is needed for the proper movement of Bassoon to the synapsis [56]. Since Bassoon is indispensable for the generation of the ribbon synapses, and the removal of its expression in the cones leads to/causes the unsynchronized release of neurotransmitters [57], GRF2 could be affecting Bassoon localization by interacting with 14-3-3γ. In addition, Bassoon phosphorylation, which is needed for its location to ribbon synapses, is mediated by p90 RSK (RSK2) [56], whose activity is modulated by the Ras signaling pathway [58]. Thus, the removal of GRF2, the main RAS activator in cone photoreceptors, could lead to the absence of RSK2 activation, Bassoon phosphorylation, and the abnormal generation of ribbon synapses in this particular cell type.

Finally, the data in this manuscript also show that GRF2 elimination leads to rapid cone cell death, which is faster on the dorsal area of the GRF2^KO^ retina (Figure 4C), a specific feature shared with many human degenerative illnesses of the retina [59]. Consistently, GRF2^KO^ mice have defective ribbon synapses, and lower levels of these synapses have also been observed in a mouse model of Bardet–Biedl syndrome (BBS), a human illness characterized by retinal dystrophy and sight loss [60]. These observations suggest that our GRF2^KO^ mice may provide a good model for studies on human diseases with primary cone cell degeneration and, due to its role in cone maintenance and function, GRF2 mutations may also be possible genetic alterations behind some forms of achromatopsia of unknown origin.

## 5. Conclusions

In this work, we show that GRF2 is a key regulator of retinal function. In addition to its previously reported role in the control of cone nuclear movement, it is also necessary for cone photoreceptor maintenance and ribbon synapse formation in adult mice. Our analyses of potential downstream target GTPases show that Rac1 is dispensable in the regulation of nuclear movement, while CDC42 has an opposed effect to GRF2, possibly through an independent signaling pathway. In addition, CDC42 cooperates with GRF2 in cone photoreceptor maintenance and ribbon synapse formation, as CDC42 elimination strongly aggravates the effects of GRF2 removal in the retina.

## Figures and Tables

**Figure 1 cells-12-02574-f001:**
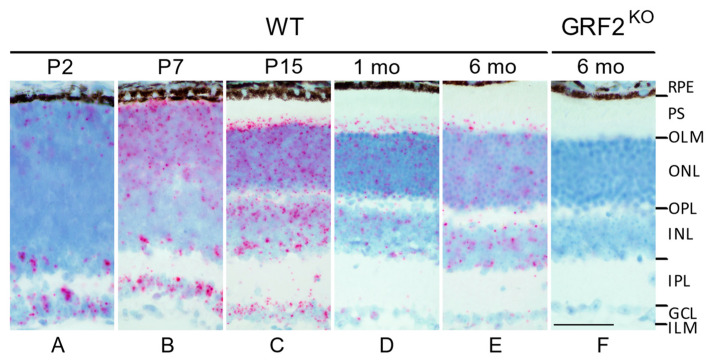
GRF2 expression in WT (**A**–**E**) and GRF2^KO^ retinas (**F**). ISH showing the expression of GRF2 in WT retinas at P2 (**A**), P7 (**B**), P15 (**C**), 1 month (**D**), and 6 months of age. (**E**) Lack of expression in the retina of a 6-month-old GRF2^KO^ mouse (**F**). Scale bar = 20 μm. RPE: retina pigmented epithelium, PS: photoreceptor segments, OLM: outer limiting membrane, ONL: outer nuclear layer, OPL: outer plexiform layer, INL: inner nuclear layer, IPL: inner plexiform layer, GCL: ganglion cell layer, ILM: inner limiting membrane.

**Figure 2 cells-12-02574-f002:**
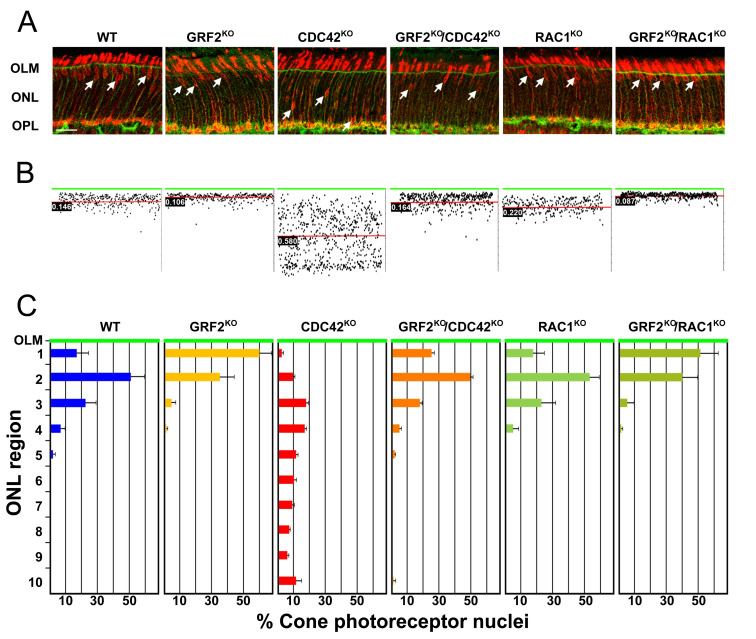
(**A**) Representative images and graphs of the cone nuclear positioning at P15 in the different genotypes analyzed. Genotypes are indicated at the top of each panel. The OLM is labeled in green with anti-β-Catenin antibodies, and cone nuclei are labeled in red with anti-R/G opsin antibodies. Arrows indicate cone nuclei. OLM = outer limiting membrane; ONL = outer nuclear layer; OPL = outer plexiform layer. (**B**) Representative graph of the cone nuclei position measured for one retina. The green line represents the OLM. The red line shows the mean position of the cone nuclei in that retina. Boxed numbers indicate the mean distance of the cone nuclei to the OLM (in mm). (**C**) Relative number of cone nuclei in each of the ten areas in which the ONL was divided. Region 1 is closest to the OLM (green line), and region 10 is the most vitreal part of the ONL (*n* = 3 for WT, RAC1^KO^, GRF2^KO^/RAC1^KO^, and GRF2^KO^/CDC42^KO^; *n* = 5 for CDC42^KO^ and GRF2^KO^.

**Figure 3 cells-12-02574-f003:**
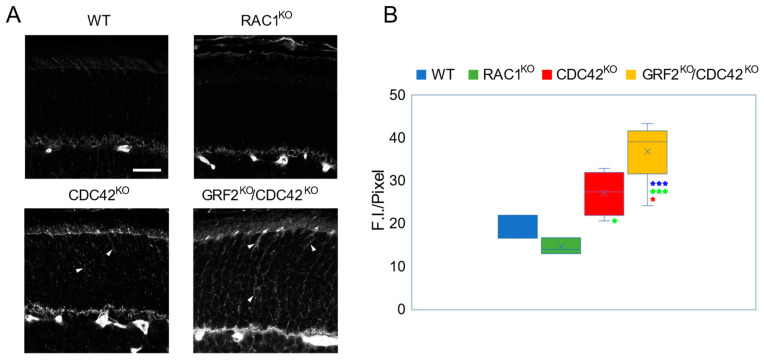
MLC2 activation in RAC1^KO^, CDC42^KO^, and GRF2^KO^/CDC42^KO^ retinas at P15. (**A**) Representative images of pMLC2 immunostaining in retinas of the indicated genotypes (arrows). Scale bar = 20 µm. (**B**) Quantification of absolute fluorescence intensity (F.I.) signal for pMLC2 immunostaining in the ONL of retinas of the four genotypes under study; *n* = 3 for WT and RAC1^KO^; *n* = 4 for CDC42^KO^; *n* = 6 for GRF2^KO^/CDC42^KO^. * *p* < 0.05, *** *p* < 0.005. Blue asterisks show significant changes against the WT samples, green asterisks refer to statistical data against Rac1^KO^. The red asterisk shows the statistical significance of the GRF2^KO^/CDC42^KO^ data against single CDC42^KO^ mice.

**Figure 4 cells-12-02574-f004:**
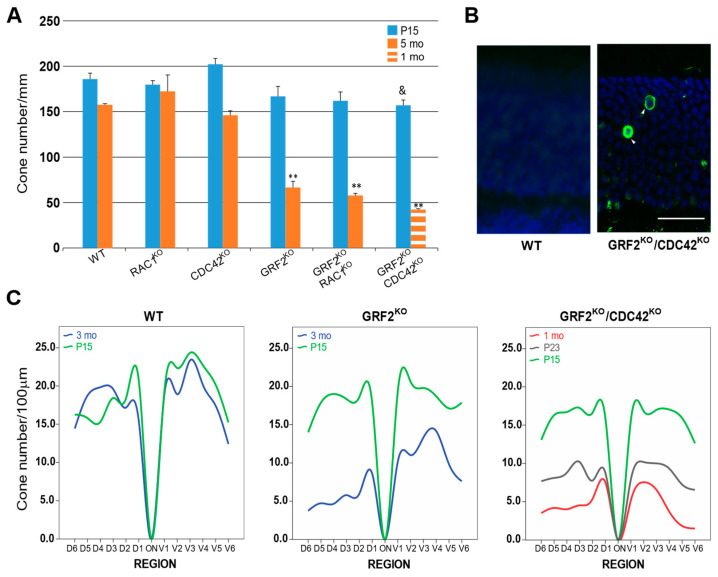
Quantification of cone cell numbers based on CRE expression. (**A**) Number of CRE+ cells/mm in whole retinal sections. The timepoints compared for all genotypes correspond to P15 (solid blue bars) and 5 months of age (solid brown bars), except for the GRF2^KO^/CDC42^KO^ mice, where only the P15 (solid blue bars), and 1-month-old timepoints (broken brown bars) were analyzed due to the rapid cone cell death occurring in these animals. (**B**) Representative TUNEL staining (arrowheads) in WT and GRF2^KO^/CDC42^KO^ P23 retinas. Scale bar = 20 µm. (**C**) Number of cone cells in 100 µm stretches of the ONL located in 6 distinct areas of the dorsal and ventral retina, at the indicated (color coded) stages of postnatal development. Due to the varying cell death rates observed in the different genotypes, the time points of the measurements shown here differ depending on the genotypes analyzed. P15 (green line) and 3 months of age (blue line) were analyzed in the WT control retinas and in the GRF2^KO^, whereas in the GRF2^KO^/CDC42^KO^, measurements were performed at P15 (green line), P23 (grey line), and 1 month of age (red line). ** *p* < 0.01 vs. WT at 5 months and *p* < 0.05 vs. WT at P15. WT: *n* = 2 for P15; *n* = 2 for 3 months old; *n* = 3 for 5 months old. RAC1^KO^: *n* = 2 for P15; *n* = 3 for 5 months old. CDC42^KO^: *n* = 3 for P15; *n* = 4 for 5 months old. GRF2^KO^: *n* = 2 for P15; *n* = 4 for 3 months old; *n* = 3 for 5 months old. GRF2^KO^/RAC1^KO^: *n* = 2 for P15; *n* = 2 for 5 months old. GRF2^KO^/CDC42^KO^: *n* = 5 for P15; *n* = 3 for P23; *n* = 2 for 1 month old.

**Figure 5 cells-12-02574-f005:**
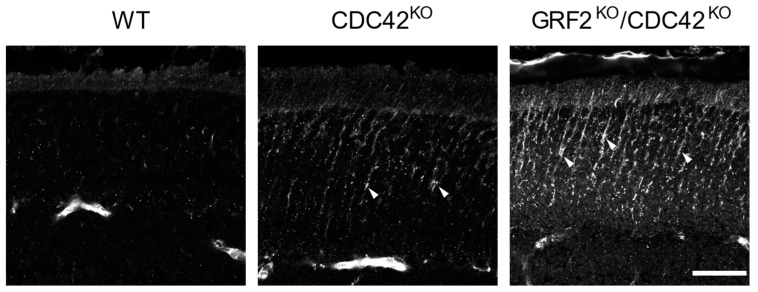
Analysis of VASP phosphorylation in WT, CDC42^KO^, and GRF2^KO^/CDC42^KO^ retinas at P15. Representative images showing VASP activation through phosphorylation, as detected using a specific pVASP antibody (arrowheads) in retinas of the indicated genotypes. Scale bar = 20 µm. WT (*n* = 3); CDC42^KO^ (*n* = 3); GRF2^KO^/CDC42^KO^ (*n* = 4).

**Figure 6 cells-12-02574-f006:**
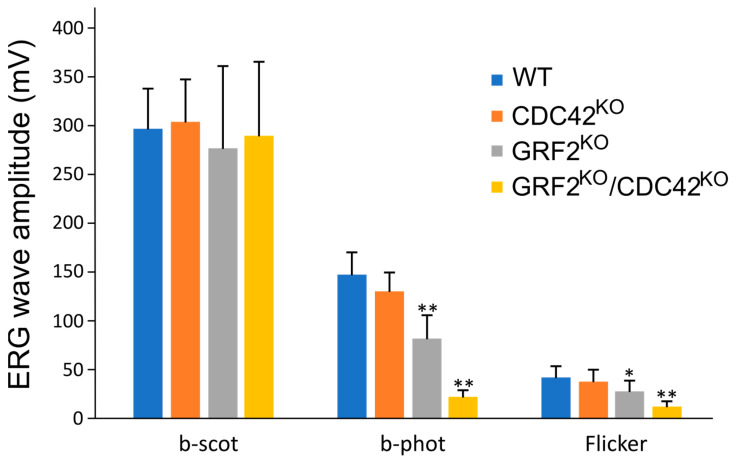
Graphical representation of the electroretinographic data recordings obtained from WT, GRF2^KO^, CDC42^KO^, and GRF2^KO^/CDC42^KO^ mice; b-scot is a measurement of rod function, while b-phot and flicker measure cone function. All recordings were performed at 1 month of age. * *p* < 0.05; ** *p* < 0.01; Student’s *t*-test with Bonferroni post hoc; WT (*n* = 7); CDC42^KO^ (*n* = 9); GRF2^KO^ (*n* = 8); GRF2^KO^/CDC42^KO^ (*n* = 7).

**Figure 7 cells-12-02574-f007:**
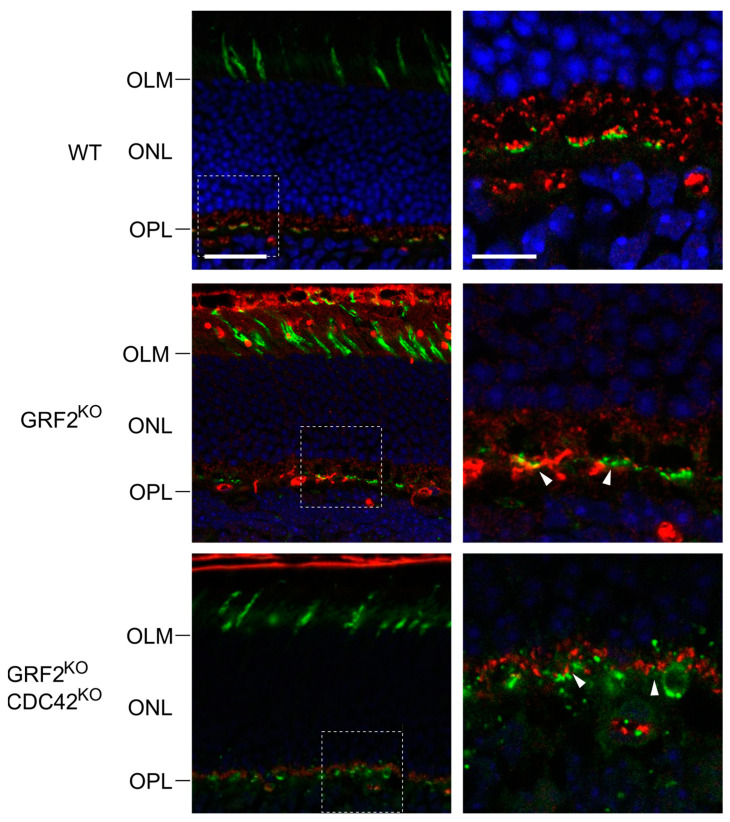
Analysis of cone photoreceptors and synaptic connections in WT, GRF2^KO^, and GRF2^KO^/CDC42^KO^ retinas. Representative retinal immunofluorescence images after immunostaining for peanut agglutinin (PNA, green) and 14-3-3γ (red). The left panels identify the position of the PS, OML, ONL, and OPL layers in retinas of the indicated genotypes. Scale bar = 30 µm. The right panels show magnifications of the indicated insets (broken line squares) corresponding to regions of synaptic interactions of the cone photoreceptors in the OPL. Arrows point to areas of misalignment between 14-3-3γ and PNA in the OPL. Scale bar = 6 µm. OLM = outer limiting membrane; ONL = outer nuclear layer; OPL = outer plexiform layer. All genotypes: *n* = 3.

**Figure 8 cells-12-02574-f008:**
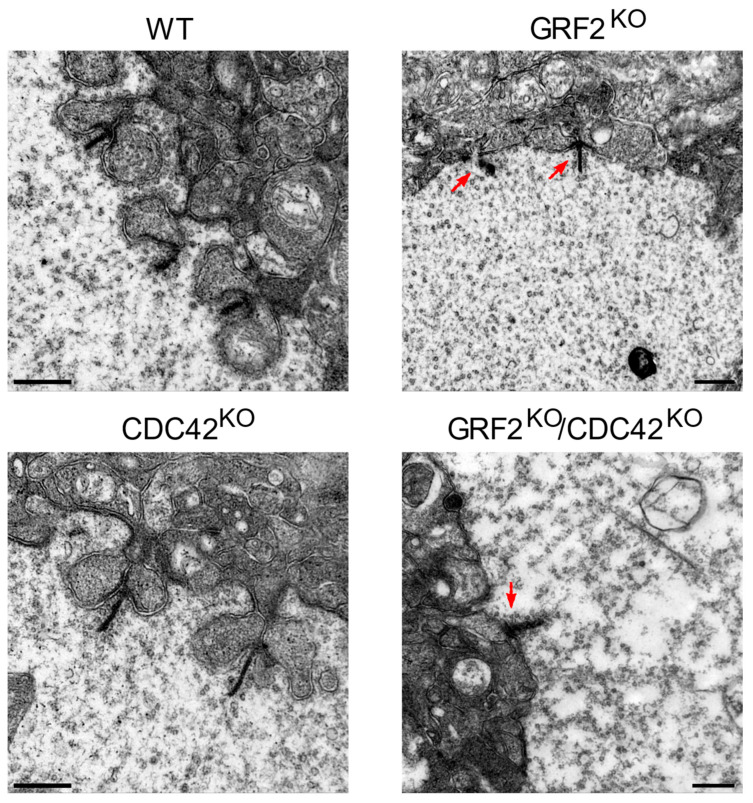
Electron microscopy analysis of ultrathin sections from WT and KO cone photoreceptor synapses. Representative electron microscopic images of the cone cell synaptic region of 1-month-old mice from WT, GRF2^KO^, CDC42^KO^, and GRF2^KO^/CDC42^KO^ mice. Scale bars = 500 nm. Arrows point to areas of specific structural alterations in the ribbon synapses. WT (*n* = 3); CDC42^KO^ (*n* = 4); GRF2^KO^ (*n* = 2); GRF2^KO^/CDC42^KO^ (*n* = 2).

**Figure 9 cells-12-02574-f009:**
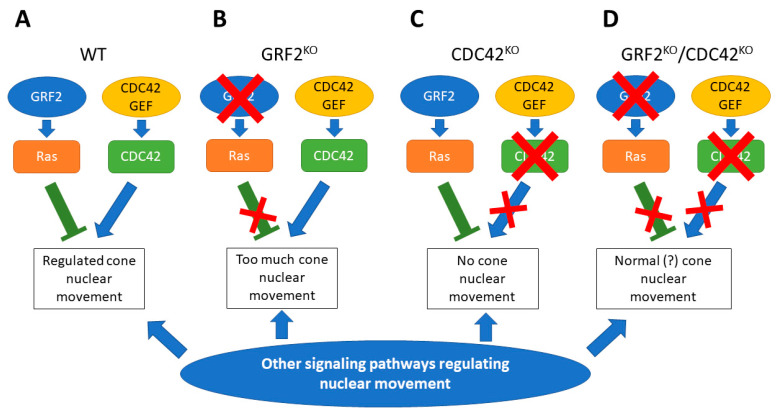
Proposed mechanistic model for the contribution of GRF2 and CDC42 to control of nuclear movement and positioning in retinal cone photoreceptors. (**A**) In the normal retina, several pathways coordinately regulate cone nuclear movement. In such a physiological context, RAS activation by GRF2 would negatively control that movement, while CDC42 activation would induce it. (**B**) GRF2 removal would lead to an excessive cone nuclear movement, with nuclei being positioned closer to the outer limiting membrane, or even surpassing it. (**C**) CDC42 elimination would lead to defective nuclear movement, with nuclei positioned at all levels in the outer nuclear layer. In this context, GRF2 presence would overcome the effect of other signaling pathways inducing nuclear movement. (**D**) When both CDC42 and GRF2 are absent, cone nuclear movement induced by alternative pathways would be enough to achieve a normal location of the nuclei at the end of the retinal development.

**Table 1 cells-12-02574-t001:** List of antibodies used in this study.

Antibody	Source	Cat#	Working Dilution
GLAST	Lifespan Biosciences (Lynnwood, WA, USA)	LS-C94136	1:500
Lectin from Arachis hypogaea-FITC	Sigma-Aldrich (St. Louis, MI, USA)	L7381	1:200
Red/Green Opsin	Chemicon (now Sigma-Aldrich, St. Louis, MI, USA)	AB5405	1:2000
ꞵ-catenin	BD Transduction Laboratories (Franklin Lakes, NJ, USA)	610154	1:500
p-MLC2	Cell Signaling (Danvers, MA, USA)	36755	1:200
p-VASP	Nanotools (Teningen, Germany)	0047-100/VASP-16C2	1:200
CRE	Millipore (Burlington, MA, USA)	MAB3120	1:1000
14-3-3γ	Upstate (Upstate, NY, USA)	05-639	1:1000
Neuron specific enolase (NSE)	Polysciences (Warrington, PA, USA)	16625	1:4000

## Data Availability

Not applicable.

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
