# Peer review of "GRF2 Is Crucial for Cone Photoreceptor Viability and Ribbon Synapse Formation in the Mouse Retina"

_cells, 2023, doi:10.3390/cells12212574_

Round 1

Reviewer 1 Report

Comments and Suggestions for Authors

This manuscript reports an interesting and accurate analysis in vivo analysis of the role of CDC42, RAC1, and GRF2 in nuclear positioning and survival of cone photoreceptors in KO and double KO mice. The manuscript is well organized and clearly written, the research, which required a large amount of effort, was well planned and rigorously executed, and the data are clearly presented and support the conclusions, which are an accurate and honest interpretation of the data.

Minor Points.

Lane 273. for better understanding, I suggest replacing the phrase with “increased levels of activated, phosphorylated MLC2 (pMLC2) in the mouse retina”

Figure 4A. In the graph, I strongly suggest adding the point at 5 months for double KO GRF2/CDC42 mouse and/or the points at 1 month at least for single KO GRF2. This will allow comparison of results obtained at the same time points in different animal models and facilitate understanding.

Fig. 5. For the same reason, I suggest adding the panel for the single GRF2- KO.

Author Response

We thank the reviewer for her/his kind comments on our work and the suggested improvements. We have used her/his suggestions to include several changes to our original manuscript:

  • Following her/his recommendation we have changed the phrasing in lane 273 (lane 281 in the revised version) for better understanding (we think that in her/his review she/he mistyped 273, because the suggested changes only make sense in the original lane 293).
  • To address the reviewer suggestion about including the data for 5 months for the DKO, we have added a new sentence commenting that at 5 months of age the DKO mice have almost no remaining cones in the retina (lines 329-331 of the revised version). We agree that the data for GRF2 KOs at 1 month of age would be useful. Unfortunately, we don’t have measurements for the GRF2 KO at that age and this GRF2 KO mice are now unavailable. Nonetheless, we believe that (i) the absence of alterations at P15 in the GRF2 KO, and, particularly (ii) the different pattern in cone cell death, (iii) the worse phenotype observed in the DKO at 1 month vs the GRF2 KO at 3 months (see Figure 4C) and (iv) the total absence of cones at 5 months of age in the DKO provide enough supporting information about the severity of cone cell death in both phenotypes.

The data for pVASP in the GRF2 retinas was already published in Jimeno et al. RASGRF2 controls nuclear migration in postnatal retinal cone photoreceptors. J Cell Sci. 2016 Feb 15;129(4):729-42. For this reason, and according to the three R’s policy using experimental animals, we didn’t repeat the experiment. We could use a representative image from those experiments different to that used in the 2016 paper, but we believe that a comparison between immunofluorescence experiments performed at different time points is not useful (The reference to the 2016 article is mentioned in lines 369 and 375). 

Reviewer 2 Report

Comments and Suggestions for Authors

The research article by Dr David Jimeno et al., entitled “GRF2 is crucial for cone photoreceptor viability and ribbon synapse formation in the mouse retina”, investigates the morphofunctional effects of the suppression of GRF2 in retinal cone photoreceptors of mice. GRF2KO and GRF2WT mice are used in this study, which shows that GRF2 is fundamental for the correct nuclear positioning in the retina, while the absence of CDC42, which is a target of GRF2, impairs the electrophysiological responses of the cone photoreceptors. Double ablation of GRF2 and CDC42 results in a severe morphofunctional impairment of cones, which also involved the ribbon synapse formation.

The study is very well designed and organized, results are shown in a very clear and accurate manner and appropriately discussed.

Some minor changes to be carried out are the following:

Abstract

1)      There are some abbreviations that are not extensively written (GEF, GRF2,…). Please provide the name in extenso of these abbreviations.

Introduction

1)      Introduction is well-written and it clearly explains the background and the purpose of the study. Some abbreviations remain to be extensively reported. If it is of their convenience and if this is accepted by the Editors/publishers, the Authors may provide a list of abbreviations.

Materials and Methods

1)      This section is written accurately. Information of materials used is complete, but the number of each genotype of mice used in the study is completely lacking. Please, provide it, and indicate how many mice are used for each procedure applied in the study (light/fluorescent microscopy, electron microscopy,…).

2)      A question arises regarding the effective abundance of cone photoreceptors in the temporal or nasal portions of the retina, where they represent a minor population compared with rods. Why the Authors did not analyze the cones within the macula? Whether distribution of cones between macular (foveal) and extramacular (extrafoveal) zones occurs at the end of the development and this concept is not applicable during the developmental phases investigated in this study, please specify it.

Results

1)      Figure 3B. In the graph, one may assume that significant differences among groups are reported with different colors. A brief explanation of the use of the colors in the legend could help the correct interpretation of data.

2)      Figure 3A. The bar is lacking.

Author Response

We thank the reviewer for her/his comments saying that our work is “very well designed and organized, results are shown in a very clear and accurate manner and appropriately discussed”. We also appreciate her/his effort in making suggestions to improve our work. Following some of her/his comments we have made several changes to our work that include:

  • A list of abbreviations has been included at the end of the manuscript. Hopefully this will comply with the editorial policy and can be admitted in the final published version of the work.
  • We have now added the number of animals used for the experiments depicted in figures 5, 7 and 8 to their corresponding figure legends. The number of animals used for the rest of the experiments were already mentioned in the respective figure legends.
  • We could not possibly analyze the cone number in the macula because, in contrast to humans, mice do not have that structure in their retinas.
  • The meaning of the colored asterisks in Figure 3B has been added to the figure legend.

A scale bar has been added to Figure 3A and the size of such bar has been added to the figure legend.
